# Respiratory Depression as Antibacterial Mechanism of Linalool against *Pseudomonas fragi* Based on Metabolomics

**DOI:** 10.3390/ijms231911586

**Published:** 2022-09-30

**Authors:** Yuansong Li, Rongrong He, Haiming Chen, Da Chen, Wenxue Chen

**Affiliations:** 1School of Precision Instrument and Opto-Electronics Engineering, Tianjin University, 92 Weijin Road, Tianjin 300072, China; 2HSF/LWL Collaborative Innovation Laboratory, School of Food Science and Engineering, Hainan University, 58 People Road, Haikou 570228, China; 3Tianjin Engineering Research Center of Civil Aviation Energy Environment and Green Development, Civil Aviation University of China, 2898 Jinbei Road, Tianjin 300300, China

**Keywords:** *Pseudomonas fragi*, linalool, antibacterial mechanism, metabolomics

## Abstract

Linalool showed a broad-spectrum antibacterial effect, but few studies have elucidated the antibacterial mechanism of linalool on *Pseudomonas fragi* (*P. fragi*) to date. The present study aimed to uncover the antimicrobial activity and potential mechanism of linalool against *P. fragi* by determining key enzyme activities and metabolites combined with a high-throughput method and metabolomic pathway analysis. As a result, linalool had excellent inhibitory activity against *P. fragi* with MIC of 1.5 mL/L. In addition, the presence of linalool significantly altered the intracellular metabolic profile and a total of 346 differential metabolites were identified, of which 201 were up-regulated and 145 were down-regulated. The highlight pathways included beta-alanine metabolism, pantothenic acid and CoA metabolism, alanine, aspartate and glutamate metabolism, nicotinate and nicotinamide metabolism. Overall, linalool could cause metabolic disorders in cells, and the main metabolic pathways involved energy metabolism, amino acid metabolism and nucleic acid metabolism. In particular, the results of intracellular ATP content and related enzymatic activities (ATPase, SDH, and GOT) also highlighted that energy limitation and amino acid disturbance occurred intracellularly. Together, these findings provided new insights into the mechanism by which linalool inhibited *P. fragi* and theoretical guidance for its development as a natural preservative.

## 1. Introduction

Meat spoilage due to bacterial growth and metabolism is a major problem in the food industry. Apart from lipid oxidation and endogenous enzymatic reactions, spoilage is mainly attributed to undesired microbial activity [1]. Refrigeration is usually the most common preservative method of meat and meat products [2]. *Pseudomonas* spp. are considered main spoilers of raw meat that has been stored under aerobic refrigerated conditions despite the diversity of bacteria was found in meat [3]. *Pseudomonas fragi* (*P. fragi*) is the most frequently found species, with an incidence between 56.7% and 79.0% on spoiled meat [4]. *P. fragi* is a facultative anaerobic and Gram-negative bacterium that is able to grow between 2 and 35 °C [5]. It has a high spoilage potential due to extracellular proteases and lipases resulting in changes in physical and organoleptic properties when acted on foods. It is worth noting that this psychrophilic species is favored by the cold chain applicable to fresh meat products [6]. Therefore, it is urgent to take effective measures against *P. fragi* within the cold chain.

In recent years, taking into account the destruction of nutrition by sterilization technology and antibiotic abuse, consumers increasingly prefer to use natural extracts as potential antibiotics to inhibit the growth of pathogenic bacteria [7]. Among them, the essential oils (EOs) of many plant species have gained momentum in many food-processing applications for controlling food-borne pathogens [8,9]. Linalool (C_10_H_18_O), also known as 3,7-dimethyl-1,6-octadien-3-ol, is a monoterpene alcohol, which is found in the EOs extracted from more than 200 plants worldwide, such as *Coriandrum sativum*, *Thymus vulgaris*, *Juniperus communis*, *Green pepper*, *Forsythia koreana leaf*, and *Pine needle* [10,11,12]. This monoterpene alcohol poses antioxidant, anti-inflammatory, and anticancer activities [13], and exhibits antibacterial activity against a variety of pathogenic bacteria. Linalool has been used in many fields due to its excellent activity. For example, it has been demonstrated that linalool inhibited glutamate-induced acute pain responses [14]. Moreover, previous literature also reported that linalool had antianxiolytic activity [15]. In industrial production, linalool was an important intermediate in the synthesis of vitamin E [12]. Regarding antibacterial aspects, many researchers believed that linalool had excellent antibacterial potential against a variety of pathogens including Pseudomonas [16,17,18]. There was evidence that linalool can exert a bacteriostatic effect by inhibiting biofilm formation and interfering with the quorum sensing system [19]. Our previous studies also found that linalool had significant antibacterial potential by damaging cell structure, enhancing cell membrane permeability and impairing respiratory metabolism [17,20,21]. Linalool is widely used in pharmaceuticals, cosmetics [22] and food additives, and approximately 1000 metric tons of linalool are consumed annually globally [12]. Therefore, the development of linalool as a natural antibacterial agent has broad application prospects.

Metabolomics analysis can detect the series of changes in metabolites caused by external stimuli in living systems [23]. Previous reports suggested that global metabolomics was an effective approach to study pharmacological mechanisms at the cellular level [24]. We also employed metabolomic techniques to track the global metabolite expression levels of *Shewanella putrefaciens* and *Listeria monocytogenes* when exposed to linalool [18,25]. However, we have not systematically elucidated the bacteriostatic activity and underlying mechanism of linalool against *P. fragi*. Therefore, the purpose of this study was to investigate the antibacterial mechanism of linalool against *P. fragi* based on metabolomics combined with enzyme activity assay. Furthermore, multivariate statistics, Kyoto Encyclopedia of Genes and Genomes (KEGG) and enzyme activity were combined to analyze the changes of metabolic pathways with or without linalool, encouraging the proposal of linalool as a natural antibacterial agent in the food industry.

## 2. Results

### 2.1. Global Analysis of Metabolomic Response

The purpose of the metabolomics approach was to characterize or quantify as many metabolites as possible in an objective manner in biological samples [26]. Metabolomics analysis also assesses external stimuli or a series of changes, and the metabolite composition is directly linked to phenotypic variations occurring in the living system. In order to analyze the level of metabolite changes in depth and the possible network pathways involved, the metabolomics profile of *P. fragi* with linalool treatment was determined by UPLC-MS/MS, which was appropriate for detecting metabolites with low molecular weight. Metabolites with only C, H, and O are detected by the negative ion mode, whereas those also containing N are ionized preferentially in the positive ion mode. Therefore, ionization should be conducted in positive and negative ways to enhance the coverage of metabolomics. Multivariate statistical analysis was performed on the identified compounds according to the data obtained in both modes to evaluate the effect of linalool treatment on metabolites of *P. fragi*. Principal component analysis (PCA) was used to reduce the multidimensional structure of the data and provide a two-dimensional graph to explain the observed differences [27]. As can been seen in Figure 1A,E, the metabolites of the treated group and the control group were separated along the PC [1] scores, and positive ion mode (ESI+) and negative ion mode (ESI−) were within the 95% confidence interval. To obtain more reliable metabolite differences between the treated group and the control group, supervised orthogonal projection to latent structures discriminant analysis (OPLS-DA) was applied to verified the whole dysregulated metabolic activities in ESI+ and ESI−, respectively (Figure 1B,F). In particular, the original model could explain the difference between the treated group and the control group. In OPLS-DA permutation test (Figure 1C,G), the intercept of the regression line of Q^2^ and the vertical axis was less than zero. Meanwhile, as the replacement retention gradually decreasing, the proportion of the replacement Y variable increased, and the Q^2^ of the random model gradually decreased, implying that the original model had good robustness and no over-fitting phenomenon in ESI+ and ESI−. The loading plot was drawn to represent the contribution of variables and potentially different substances. PCA loading plots of two modes were shown in Figure 1D,H. Additionally, the metabolites were aggregated with good reproducibility, specifically.

### 2.2. Cluster Analysis of Differential Metabolites

Differential metabolites were identified and plotted as heat maps (Figure 2A and Figure 3A) between the linalool-treated group and the control group. In addition, the first principal component of VIP (Variable Importance in the Projection) >1 and *p* < 0.05 (Student’s *t*-test) were confirmed as differential metabolites [28]. In ESI+, a total of 194 differential metabolites were identified, among which 107 were up-regulated and 87 were down-regulated. In ESI−, there were 152 differential metabolites, among which 58 were down-regulated, and 94 were significantly up-regulated. The selected differential metabolites were visualized in the form of a volcano plot and then plotted (Figure 2B and Figure 3B). The metabolites that were significantly up-regulated are shown in red, those significantly down regulated are shown in blue, and metabolites that were not significantly different are in gray. Visualization of disturbed metabolic pathways in linalool-treated strain were shown in Figure 2C and Figure 3C. The disturbed metabolites were annotated into different pathway through KEGG enrichment analysis, and results (ESI+) showed that the majority of differential metabolites were related to beta-alanine metabolism, pantothenic acid and CoA biosynthesis, alanine, aspartate and glutamate metabolism, nicotinate and nicotinamide metabolism, D-glutamate and D-glutamine metabolism. In ESI−, pantothenic acid and CoA biosynthesis and alanine, aspartate and glutamate metabolism were enriched.

### 2.3. Intracellular Enzyme Activity (SDH, ATPase and GOT) and Composition (ATP) Analysis

As shown in Figure 4A, the intracellular ATP content showed a significant decreasing trend after adding linalool compared with the control group. This meant that there may be a lack of energy metabolism in cells under the stress of linalool. Intracellular activities of two key enzymes related to energy metabolism were also concerned in the current study. As shown in Figure 4B,C, the activities of SDH and ATPase were significantly inhibited in the presence of linalool, which also supported the conclusion of disordered-energy metabolism. These data revealed that intracellular energy limitation may be one of the lethal mechanisms of linalool against *P. fragi*. Aspartate aminotransferase/aspartate aminotransferase activity was also found to be significantly reduced (Figure 4D), which indicated that the conversion among some amino acids was inhibited and thus might further affect the amino acid metabolism.

## 3. Discussion

Meat, as a good source of protein for human intake, often faces the challenge of spoilage due to various nutrients. Refrigeration is usually the most common preservative method of meat and its processed products [2]. However, the combination of a long storage time and a low temperature creates a selective advantage for psychrotrophic bacteria, especially *Pseudomonas* spp, which plays a major role in limiting the shelf life of the chilled meat [3]. There was evidence that linalool can exert an anti- *Acinetobacter baumannii* effect by inhibiting biofilm formation and interfering with the quorum sensing system [10]. Our previous research also confirmed that linalool showed inhibitory effects on a variety of pathogenic bacteria that caused food spoilage, including *Pseudomonas aeruginosa*, *Shewanella putrefaciens* [20,25]. In the present study, the mechanism of linalool inhibiting *P. fragi* was further explored based on metabolomics combined with key enzyme activity assay.

Omics analysis, as a powerful technique to identify differential metabolites, expressed genes and proteins, is widely used [29]. Metabolomics technology can detect the dynamic changes of the metabolite spectrum based on the organism’s response to external stimuli [30]. Different metabolites identified in the two modes (ESI+ and ESI−) were introduced into KEGG for enrichment pathway analysis. The main pathways were grouped into the following categories and discussed with emphasis in the following work, namely energy metabolism, amino acid metabolism and nucleic acid metabolism.

(i) Energy metabolism. Under the action of linalool, a variety of intracellular energy-related pathways were all affected as showed in Figure 5. For bacteria, various intracellular physiological activities including material transport, cytoskeleton shaping and catalytic reactions require energy to drive. In this study, intracellular energy metabolism-related pathways including pentose phosphate pathway, pentose and glucuronic acid interconversion, galactose metabolism, glycerolipid metabolism, glycerophospholipid metabolism, TCA, pantothenic acid and CoA biosynthesis and pyruvate metabolism were all activated. These data supported that energy metabolism was disrupted under linalool stress, similar to our previous report [18,25]. D-fructose 6-phosphate was converted to fructose 1,6-bisphosphate by consuming ATP under the catalysis of phosphofructokinase, the first committed step of glycolysis, which is generally inhibited by a high concentration of citric acid and ATP [31]. However, citric acid level was found to increase after exposure to linalool in the current study. This derivative effect was likely to trigger disturbances in the glycolytic pathway although it was not currently enriched for this pathway. In addition, citrate was involved in the TCA cycle, suggesting that the intracellular central pathway was also disturbed. The TCA cycle was the major central pathway linking the majority of individual metabolic pathways, and it provided precursors for many biosynthetic pathways due to its function in the oxidative catabolism of carbohydrates, fatty acids and amino acids. Unfortunately, citric acid was also involved in glyoxylate and dicarboxylic acid metabolism, and L-glutamate, another player in this pathway, was also up-regulated. As the precursor of CoA and the repairing base for acyl carrier protein (ACP), and participates in carbohydrate, fatty acid, protein and energy metabolism in living organisms, pantothenic acid is a key intermediate of pantothenate and CoA biosynthesis. The important metabolites involved in this pathway including pantothenic acid, 5,6-dihydrouracil, 3-ureidopropionic acid and uracil were all overexpressed. The CoA biosynthetic pathway became a popular target for the development of novel antimicrobial agents due to its idiographic divergences in the catalysis of the PanK reaction and adenosine transferase (PPAT) in bacterial and mammalian proteins [32]. In the pentose phosphate pathway, the contents of D-ribose, D-gluconic acid, alpha-D-ribose 1-P and D-xylulose were all up-regulated, which implied that the pathways were activated under linalool treatment. In addition to helping to scavenge reactive oxygen species by promoting the regeneration of the reducing agent nicotinamide adenine dinucleotide phosphate (NADPH) [33], the pentose phosphate pathway (PPP) can also produce 5-phosphate ribose, the precursor substance of nucleotides, which is essential for maintaining carbon steady state and nucleotide and amino acid biosynthesis [34]. Two other important energy-producing pathways, glycerolipid metabolism and glycerophospholipid metabolism, were also affected due to the detection of sn-glycero-3-P showing an upward trend. Therefore, disturbance of intracellular energy metabolism are fatal to bacteria. All of the above results suggested that linalool caused a disturbance of important energy metabolism in *P. fragi*.

(ii) Amino acid metabolism. After linalool treatment, intracellular amino acid-related metabolism suffered much more change than other pathways, which was supported by result from bubble charts (Figure 2C and Figure 3C). In amino acid anabolism (Figure 5), mainly involved in arginine biosynthesis, aminoacyl-tRNA biosynthesis, valine, leucine and isoleucine biosynthesis, lysine biosynthesis, phenylalanine, tyrosine and tryptophan biosynthesis. L-glutamic acid, N-acetylornithine, L-citrulline, and L-aspartic acid were detected, showing an upward trend in arginine biosynthesis. Regarding aminoacyl-tRNA metabolism, the contents of its important players L-histidine, L-phenylalanine, L-glutamic acid, L-lysine and L-aspartic acid were significantly increased in the presence of linalool. 2-Isopropylmalate, L-lysine and L-phenylalanine were detected to be overexpressed, respectively in valine, leucine and isoleucine biosynthesis, lysine biosynthesis, and phenylalanine, tyrosine and tryptophan biosynthesis. These data suggested that an imbalance in amino acid metabolism occurred after linalool treatment. Nevertheless, these amino acids play an important role in the physiological metabolism of bacteria. Lysine, a major component of peptidoglycan, was up-regulated, a potential explanation may be that the cells lost the ability to remodel the cytoskeleton resulting in an excess of L-lysine [35]. Glutamic acid (down-regulation) is critical for maintaining the redox reaction balance and avoid oxidative stress by producing glutathione [36]. In particular, glutathione metabolism was also disturbed in the current study, which was able to speculate that linalool may have caused oxidative stress in *P. fragi.* As an important part of protein translation, the aminoacyl- tRNA synthesis pathway was disturbed due to the detection of some amino acids (L-histidine, L-phenylalanine, L-aspartic acid, L-lysine and L-glutamic acid). It was possible that the decrease in the translation rate of intracellular proteins caused the increase in the content of these amino acids. At the same time, histidine metabolism, D-glutamine and D-glutamate metabolism, β-alanine metabolism, phenylalanine metabolism, alanine, aspartate and glutamate metabolism, cysteine and methionine metabolism, arginine and proline metabolism, lysine degradation, glycine, serine and threonine metabolism, and tryptophan metabolism were also disrupted when exposed to linalool. Moreover, the amino acids involved all showed an upward trend. In conclusion, the reason why both the amino acid biosynthesis pathway and the metabolic pathway were activated may be the adjustment made by bacteria to adapt to the unfavorable environment [18].

(iii) Nucleic acid metabolism. Linalool treatment also caused disturbances in intracellular nucleic acid metabolism, and the results were presented in Figure 5. In purine metabolism, xanthine, adenosine, alpha-D-ribose-1-P, uric acid, AMP and hypoxanthine were overexpressed, while adenine and guanosine were inhibited. As for pyrimidine metabolism, we found that the contents of thymidine, orotate, deoxyuridine, 5,6-dihydrouracil, 3-ureidopropionate, cytosine, uracil and thymine all showed an upward trend. As a raw material for purine synthesis, the content of hypoxanthine increased significantly. It is speculated that the synthesis rate of the adenine may be slowed down due to the detected decrease in the adenine content [37].

Based on the metabolome results, the content of ATP, an intracellular energy carrier, was determined. It was found that the content of ATP showed a clear downward trend under linalool stress. To clarify the mechanism behind, the activity of ATPase was measured. Among multiple enzymes that can promote the production and metabolism of ATP, ATPase is one of the essential enzymes for this mechanism [38]. The results confirmed that the activity of ATPase was inhibited in the presence of linalool. The downward of intracellular ATP content can be explained by the following mechanisms: On the one hand, ATP leakage caused by increased cellular trend permeability and ATP dissipation caused by increased proton pump hydrolysis rate, consistent with our previous research [16]. On the other hand, the main enzymes that catalyze ATP production activity was inhibited and its production rate decreased. The activity of SDH was also measured to clarify the cause of energy limitation, and it was found that the activity of SDH was also significantly inhibited by linalool. SDH plays a crucial role in the TCA cycle and in the complex II of the mitochondrial electron transport chain and is responsible for transferring electrons from succinate to ubiquinone (coenzymeQ) [31]. The reason for the inhibition of the activity of SDH may be that it is located on the cell membrane, and the damage to the cell membrane is likely to cause its activity to be inhibited. Previous studies demonstrated that linalool could damage bacterial cell structure and enhance the cell membrane permeability [39]. The present data supported that GOT activity was also inhibited, which implied that interconversions between specific amino acids were affected, consistent with the perturbed amino acid metabolism mentioned in metabolomics.

## 4. Materials and Methods

### 4.1. Materials and Bacterial Culture

*Pseudomonas fragi* (*P. fragi*) CGMCC1.3349 was purchased from China General Microbiological Culture Collection Center (Beijing, China). *P. fragi* was cultured on a nutrient agar medium (NA) (Beijing Solarbio Science & Technology Co., Ltd., Beijing, China) at 30 °C for 24 h and stored at −80 °C in sterile glycerin. Linalool (≥99%) was purchased from the Hainan Camphora Biotech Co., Ltd. (Haikou, China). Enzymes including succinate dehydrogenase (SDH), ATPase and aspartate aminotransferase/aspartate aminotransferase (GOT), intracellular components (ATP) were measured according to assay kits manual from Nanjing JianCheng Bioengineering Institute (Nanjing, China).

### 4.2. Intracellular Enzyme Activity and Composition Determination

Intracellular enzyme activity and metabolite content were determined under linalool stress as described in previous studies and the concentration of linalool referred to our previous report [39,40]. In short, linalool (MIC and 2MIC) was added into cell culture solution in the logarithmic growth phase for co-cultivation, and equal amounts of sterile water and ethanol (1%, *v*/*v*) were used as blank group and negative control group. The samples were centrifuged (6000 r/min, 10 min) to gather precipitation, then washed 3 times with sterile PBS (0.1 mol/L, pH7.4) and resuspended. The prepared bacteria cells were processed with ultrasound (200 W, interval 10 s, 30 times) to release intracellular contents and enzymes including SDH, ATP, GOT and ATPase, and then the supernatant was obtained and stored at 4 °C after centrifugation. Enzyme activity and metabolites were measured according to the kit instructions.

### 4.3. Metabolite Sample Preparation and Metabolomics Analysis

#### 4.3.1. Sample Preparation

The activated *P. fragi* was inoculated into a nutrient broth medium (NB) and cultured to the logarithmic growth phase. Then, linalool was added for co-cultivation, while an equal amount of sterile water was used as a control. The bacterial suspension was collected by centrifugation (6000 r/min, 10 min) and washed 3 times with sterile PBS. After centrifugation, bacterial cells were quickly quenched in liquid nitrogen and stored at −80 °C for analysis. The experimental samples were divided into a treated group and a control group, and six biological replicates were prepared for each group.

#### 4.3.2. Extraction of the Intracellular Metabolites

Methanol (400 μL, 80%, *v*/*v*) was added to the sample after thawing slowly on ice. The extract was obtained by ultrasound for 30 min before standing for 1 h at −20 °C. After that, the extraction solution (200 μL) was mixed with the internal standard (5 μL, 1 mg/mL dichlorophenylalanine) and transferred to a bottle for analysis after centrifugation (4 °C, 12,000 r/min, 15 min).

#### 4.3.3. UPLC-MS/MS Analysis

UPLC-MS/MS was used to analyze the changes in intracellular metabolism levels in the presence or absence of linalool with reference to previous reports [18]. The metabolites were quantified using a UPLC (Thermo, Waltham, MA, USA, Q Exactive) coupled to column (Acquity UPLC HSS T3, 2.1 × 100 mm, 1.8 μm) equipped with a quadrupole time-of-flight mass spectrometer with an electrospray ionization (ESI) source operating in positive and negative ion modes. Specifically, the sample was eluted on the column with gradient elution separation solution contained buffer A (water containing 0.05% formic acid) and mobile phase B (acetonitrile) at 0.3 mL/min. During the entire analysis, the sample was placed in the autosampler at 4 °C. The following gradients were used for separation: T = 0–1 min, 5%B, ramped linearly to 95% B at 12 min and maintained at 95% B until 13.5 min, 5%B at 13.6 min and maintained at 5%B until 16 min. Column temperature was 40 °C and sample volume was 2 μL. The ESI were performed under the conditions: ESI+: Heater Temp 300 °C; Sheath Gas Flow rate, 45 arb; Aux Gas Flow Rate, 15 arb; Sweep Gas Flow Rate, 1arb; spray voltage, 3.0 kV; Capillary Temp, 350 °C; S-Lens RF Level, 30%. ESI−: Heater Temp 300 °C, Sheath Gas Flow rate, 45arb; Aux Gas Flow Rate, 15arb; Sweep Gas Flow Rate, 1arb; spray voltage, 3.2 kV; Capillary Temp, 350 °C; S-Lens RF Level, 60%.

The data was performed feature extraction and preprocessed with Compound Discoverer software (Thermo), and then normalized and edited into two-dimensional data matrix by Microsoft Excel 2016 software, including Retention time (RT), Compound Molecular Weight (compMW), Observations (samples), number of extractable substances (ID), and peak intensity.

### 4.4. Statistical Analysis

Each experiment was repeated 3 times individually. Results were expressed as mean ± standard deviation (SD) by Origin Pro. 2018 software (Origin Lab Co., Northampton, MA, USA). The significance of difference (*p* < 0.05) between means was performed by one-way ANOVA and Duncan’s multiple range tests with SPSS version 22.0 statistical software (SPSS Inc., Chicago, IL, USA).

## 5. Conclusions

In order to fill the gap on the inhibition of *P. fragi* by linalool and deal with the challenge of pathogenic bacteria contamination in the food industry, UPLC-MS/MS-based metabolomics was used to systematically investigate the inhibitory effect of linalool on *P. fragi* and provided a novel understanding of the metabolic changes in *P. fragi* during treatment with linalool. The results showed that linalool had outstanding inhibitory activity against *P. fragi* with MIC of 1.5 mL/L (Table 1). In addition, multiple intracellular metabolites were differentially expressed including citric acid, D-ribose and D-gluconic acid after linalool treatment. KEGG analysis found that linalool can cause intracellular metabolic disorders, and the prominent metabolic pathways involved energy metabolism, amino acid metabolism and nucleic acid metabolism. The results for intracellular ATP content and related enzyme activities (ATPase, SDH and GOT) also indicated that intracellular energy limitation and amino acid disorder occurred, considered the main mechanism by which linalool inhibited *P. fragi*. All these provided a novel understanding of the directions of the metabolic changes in the inhibition of bacteria by linalool at the molecular level, which were instructive for the control of food contamination caused by *P. fragi*.

## Figures and Tables

**Figure 1 ijms-23-11586-f001:**
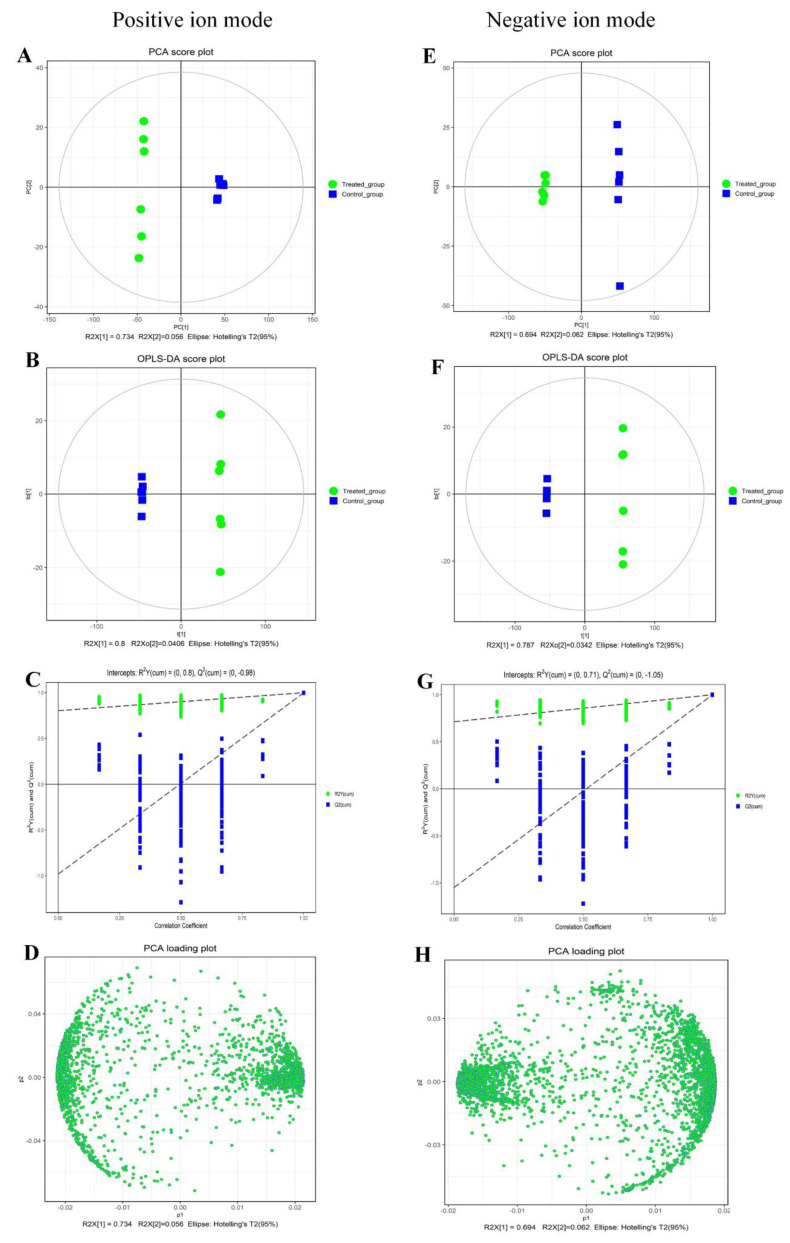
Multivariate cluster analyses of metabolite profiles of *P. fragi* with or without linalool. PCA score plot (**A**,**E**); OPLS-DA score plot (**B**,**F**); OPLS-DA permutation plot (**C**,**G**); PCA loading plot (**D**,**H**). Six biological replicates were executed to ensure reliability.

**Figure 2 ijms-23-11586-f002:**
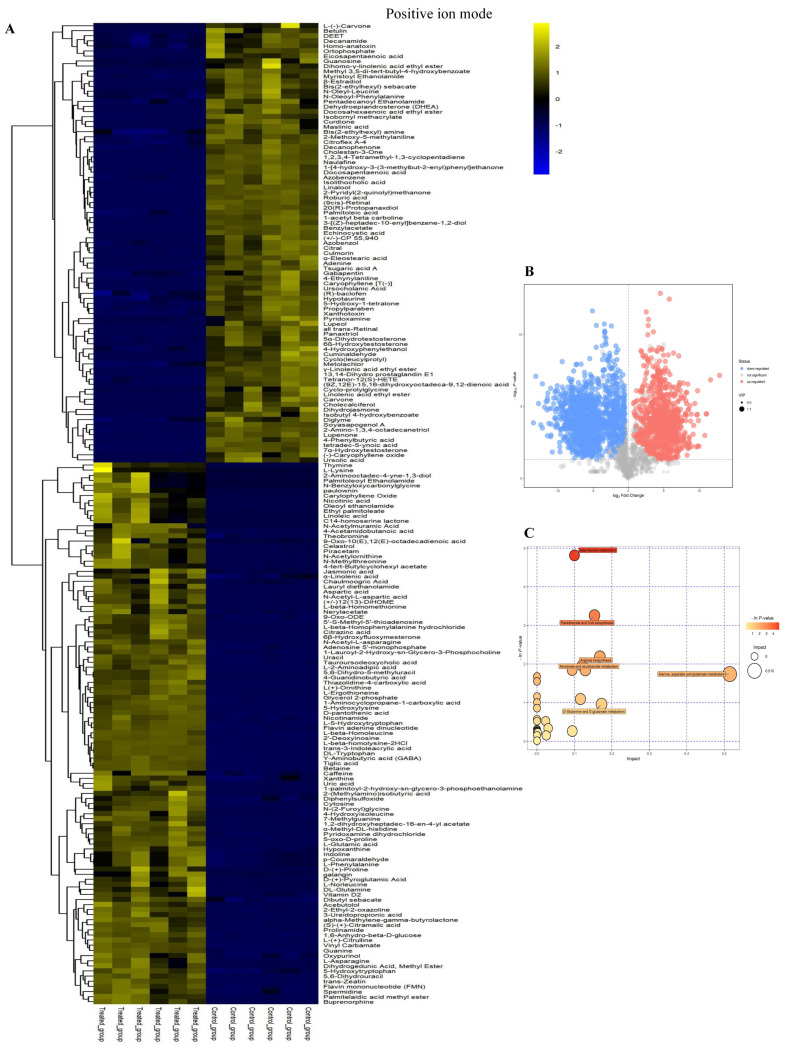
Metabolic profiles analyses of *P. fragi* between linalool-treated and control in positive ion mode. Heat maps of differential metabolites (**A**). Volcano plot (**B**). Each point in the volcanic map represents a metabolite. impacted metabolic pathway with or without linalool (**C**).

**Figure 3 ijms-23-11586-f003:**
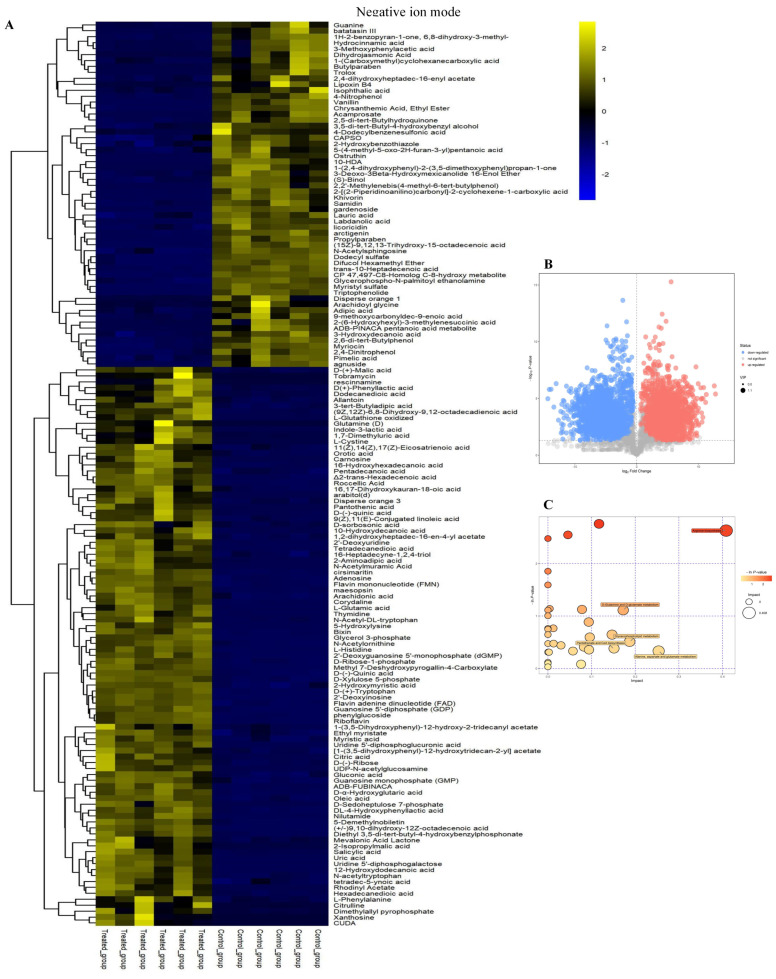
Metabolic profiles analyses of *P. fragi* between linalool-treated and control in negative ion mode. Heat maps of differential metabolites (**A**). Volcano plot (**B**). Each point in the volcanic map represents a metabolite. impacted metabolic pathways with or without linalool (**C**).

**Figure 4 ijms-23-11586-f004:**
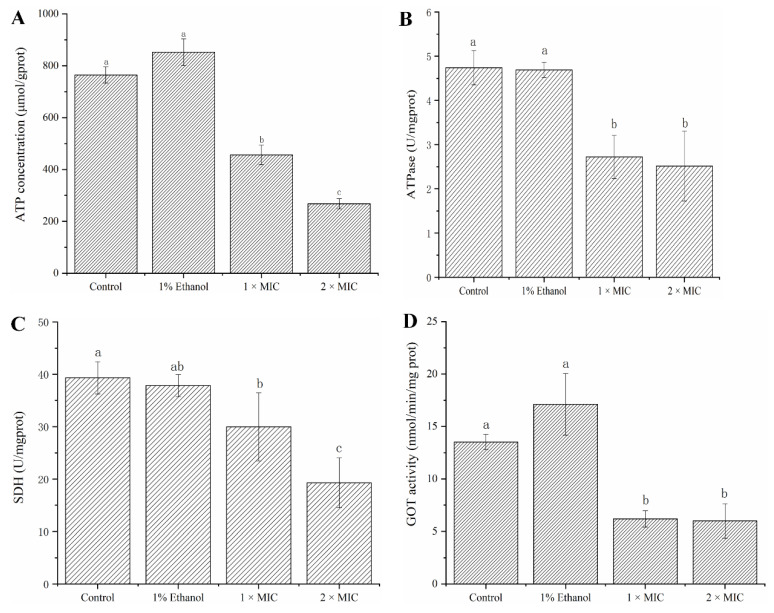
ATP content (**A**); the activity of enzymes including ATPase (**B**), SDH (**C**) and GOT (**D**). Different letters represented differences between treatments.

**Figure 5 ijms-23-11586-f005:**
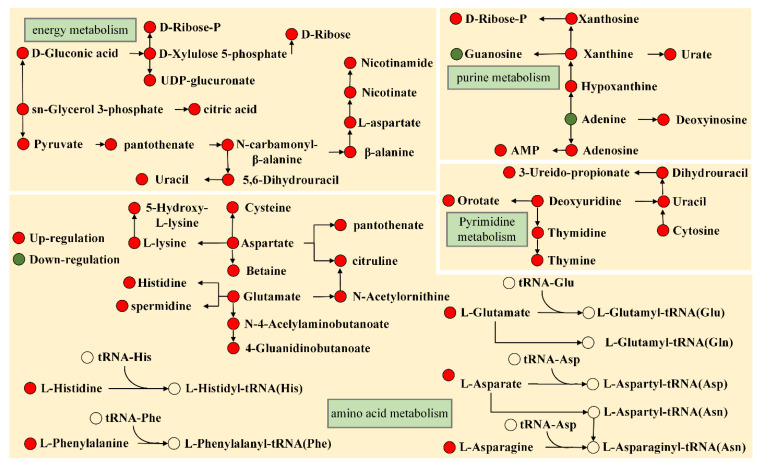
Pathway analysis of differential metabolites related to Amino acid metabolism; Energy Metabolism; Nucleic acid metabolism.

**Table 1 ijms-23-11586-t001:** Minimum inhibitory concentrations (MICs) of linalool against *P. fragi*.

Strain	Control		The Concentration of Inhibitor (mL/L)				
*P. fragi*	Water	1% Ethanol	0.375	0.75	1.5	3	6
	+++	+++	+++	++	-	-	-

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
