# Peer review of "Respiratory Depression as Antibacterial Mechanism of Linalool against Pseudomonas fragi Based on Metabolomics"

_ijms, 2022, doi:10.3390/ijms231911586_

Round 1
Reviewer 1 Report
This m/s reports the establishment of the mode of action for antibacterial activity of linalool using metabolomics; it was found that inhibition of energy and amino acid metabolism were principally affected. This paper is clearly written (although needs some correction to english language) and the results are of importance for the area. Publication is recommended.
Author Response
Thanks for your carful comment.
Reviewer 2 Report
Comments to the authors:
1. The references should be present by closely following the guidelines of the journal IJMS.
2. In line 51 and 52: The plant species names should be in italic
3. In the section Discussion, lines 233-242: the content is repeated from the section Introduction, therefore it is suggestible to modify or remove.
Author Response
Question: The references should be present by closely following the guidelines of the journal IJMS.
Answer: The format of the references in this manuscript has been revised closely following the guidelines of the journal IJMS.
Question: In line 51 and 52: The plant species names should be in italic
Answer: It has been revised according to your suggestion (lines 49-50).
Question: In the section Discussion, lines 233-242: the content is repeated from the section Introduction, therefore it is suggestible to modify or remove.
Answer: It has been removed according to your suggestion (lines 221-232).
Reviewer 3 Report
Dear authors,
Manuscript ijms-1911396 entiteled "Antibacterial mechanism of linalool against Pseudomonas fragi based on metabolomics" ands authored by : Yuansong Li , Rongrong He , Haiming Chen * , Da Chen , Wenxue Chen" targets a hot topic that is potentially very interesting for the journal readers and the scientific community. The findings are sound and really relevant to the field.
It was pleasure to me to read your manuscript. I appreciated a lot your findings and the relevance of your conclusions. I am really very supportive of the publication of your manuscript that I hope is the start point for unravelling the mode of action of linalool. Unfortunately I could not recommand your manuscript for publication at this stage in fact few points have to be addressed to provide the journal readers with a highly relevant data:
1. I do not the title relevant to the manuscript. Please highlight your findings I really think that changing the title can encourage readers to go deep in it.
2. Abstract is very weak and non informative please highlight your findings. Personally your abstract will not encourage me to read the paper if I wasn't a reviewer.
3. In the introduction please report widely the different fields where linalool have been reported. In fact linalool have been widely reported in the field of plant pathogen interactions where it is mentioned without any proof that linalool could be an antibacterial agent against numerous Pseudomonas and other bacterial pathogens. In this field your findings are highly relevant and I am sure that your results will be highly appreciated by plant pathogen background journal readers.
4. It is not possible to publish your manuscript without improving the redability of your figures. Please address this point if you want your paper useful to the journal readers.
5. I was chocked reading the conclusion. It is pitty in my point of view that you not highlight your findings and putative applications of your findings. Please identify gaps in the state of research in the field and highlight urgent research directions in the field to encourage debate in this research topic.
I will be happy to read an improved version of your manuscript that I can recommand for publication.
Best regards
Author Response
Question: I do not the title relevant to the manuscript. Please highlight your findings I really think that changing the title can encourage readers to go deep in it.
Answer: The title has been modified according to your suggestion and marked in red (lines 2-3).
Question: Abstract is very weak and non informative please highlight your findings. Personally your abstract will not encourage me to read the paper if I wasn't a reviewer.
Answer: Relevant findings in this study have been supplemented in the abstract based on your suggestions and marked in red (lines 16-21).
Question: In the introduction please report widely the different fields where linalool have been reported. In fact linalool have been widely reported in the field of plant pathogen interactions where it is mentioned without any proof that linalool could be an antibacterial agent against numerous Pseudomonas and other bacterial pathogens. In this field your findings are highly relevant and I am sure that your results will be highly appreciated by plant pathogen background journal readers.
Answer: The application of linalool in different fields has been added in this manuscript according to your suggestion and marked in red (lines 52-56). The report on the inhibitory effect of linalool on pathogenic bacteria especially Pseudomonas has been supplemented in this manuscript and marked in red (lines 56-58).
Question: It is not possible to publish your manuscript without improving the redability of your figures. Please address this point if you want your paper useful to the journal readers.
Answer: The readability of the figures in the manuscript has been improved as much as possible according to your suggestions.
Question: I was chocked reading the conclusion. It is pitty in my point of view that you not highlight your findings and putative applications of your findings. Please identify gaps in the state of research in the field and highlight urgent research directions in the field to encourage debate in this research topic.
Answer: The findings and putative applications of this study have been supplemented in the conclusion (lines 350-353, 355-360). The gaps and urgency of research in this area have been described in the current manuscript based on your suggestion (lines 346-347).
Thanks again for your careful work.
Round 2
Reviewer 3 Report
Dear authors,
Thanks for addressing my comments. The manuscript is now improved and I recommend its publication.
Best regards